# ZIF-67 Incorporated Sulfonated Poly (Aryl Ether Sulfone) Mixed Matrix Membranes for Pervaporation Separation of Methanol/Methyl Tert-Butyl Ether Mixture

**DOI:** 10.3390/membranes13040389

**Published:** 2023-03-29

**Authors:** Guanglu Han, Jie Lv, Mohan Chen

**Affiliations:** 1School of Materials and Chemical Engineering, Zhengzhou University of Light Industry, Zhengzhou 450001, China; 2Henan Engineering Research Center of Chemical Engineering Separation Process Intensification, Zhengzhou 450001, China; lvjie19850112@163.com (J.L.); chenmohan0610@163.com (M.C.)

**Keywords:** separation, pervaporation, mixed matrix membrane, ZIF-67, sulfonated poly (aryl ether sulfone)

## Abstract

Mixed matrix membranes (MMMs) with nano-fillers dispersed in polymer matrix have been proposed as alternative pervaporation membrane materials. They possess both promising selectivity benefiting from the fillers and economical processing capabilities of polymers. ZIF-67 was synthesized and incorporated into the sulfonated poly (aryl ether sulfone) (SPES) matrix to prepare SPES/ZIF-67 mixed matrix membranes with different ZIF-67 mass fractions. The as-prepared membranes were used for pervaporation separation of methanol/methyl tert-butyl ether mixtures. X-ray diffraction (XRD), Scanning Electron Microscopy (SEM) and laser particle size analysis results show that ZIF-67 is successfully synthesized, and the particle size is mainly between 280 nm and 400 nm. The membranes were characterized by SEM, atomic force microscope (AFM), water contact angle, thermogravimetric analysis (TGA), mechanical property testing and positron annihilation technique (PAT), sorption and swelling experiments, and the pervaporation performance was also investigated. The results reveal that ZIF-67 particles disperse uniformly in the SPES matrix. The roughness and hydrophilicity are enhanced by ZIF-67 exposed on the membrane surface. The mixed matrix membrane has good thermal stability and mechanical properties, which can meet the requirements of pervaporation operation. The introduction of ZIF-67 effectively regulates the free volume parameters of the mixed matrix membrane. With increasing ZIF-67 mass fraction, the cavity radius and free volume fraction increase gradually. When the operating temperature is 40 °C, the flow rate is 50 L·h^−1^ and the mass fraction of methanol in feed is 15%, the mixed matrix membrane with ZIF-67 mass fraction of 20% shows the best comprehensive pervaporation performance. The total flux and separation factor reach 0.297 kg·m^−2^·h^−1^ and 2123, respectively.

## 1. Introduction

Methanol (MeOH)/methyl tert-butyl ether (MTBE) mixture is one typical organic mixture which is obtained in the process of producing MTBE by reaction of MeOH with isobutylene. In the production process, excessive MeOH is added into the reaction system to improve the reaction conversion ratio, which encounters a problem in purification process. In addition, MeOH and MTBE form an azeotrope consisting of 14.3 wt% MeOH at atmospheric pressure. MeOH/MTBE mixture is often pretreated by water washing, followed by further separation via distillation to recycle MeOH back to the reactor. However, this conventional process is both expensive and energy intensive.

Pervaporation (PV) technology, due to its efficiency and economic benefits, has been widely and deeply studied in the fields of organic solvent dehydration [1,2,3,4,5], organic removal from water [6,7,8,9,10,11], and organic mixture separation [12,13,14,15]. In pervaporation process, feed liquid comes in contact with membrane surface. Separation takes place by preferential sorption and diffusion of the desired component through a dense membrane. Therefore, the chemical and physical structure of membranes often has a significant influence on membrane performance. PV units are smaller than conventional distillation units, and therefore have smaller footprints. With decreasing the ratio of equipment size over production capacity, PV meets the requirements of process intensification.

In the field of organic separation, the core of pervaporation technology is to design and prepare membrane materials according to the characteristics of organic mixture system. Among many membrane materials, the mixed matrix membranes (MMMs) obtained by introducing nano-fillers into the polymer matrix is one of the most important membrane materials [16,17,18]. Over the past decades, a plethora of nanostructured materials including carbon nanoparticles have been used as fillers in the polymer matrix to fabricate nanocomposite membranes with higher PV separation performance [19]. The filler contains carbon nanoparticles [20,21,22,23,24], zeolites [25,26,27,28,29], metal oxides [30,31,32,33,34,35,36], nano-clay [37,38,39], metal-organic frameworks (MOFs) [40,41,42,43,44] and so on. The concept of metal–organic frame materials (MOFs) was first proposed by American scholar Yaghi et al. in 1995 [45]. MOFs are porous structures that are composed of metal ions and polydentate organic molecules combined into a three-dimensional framework through strong metal–ligand interactions. The advantages of MOFs, such as high specific surface area and porosity, adjustable pore size, and easy functional modification, have promoted extensive research in many fields such as separation, catalysis, optical devices, electromagnetic materials, drug release, and molecular recognition. The introduction of MOFs materials plays an active role in changing the arrangement and stacking mode of polymer chains, regulating the spacing of polymer chains, optimizing the free volume parameters of membranes [46]. Thus, the separation performance is significantly improved. At the same time, organic ligands in MOFs materials can improve their compatibility with polymer matrix and thus avoid membrane defects [41,42].

Poly (aryl ether sulfone) (PES) is a novel kind of thermo-plastics engineering material. Its glass transition temperature reaches 260 °C, which makes it a thermally stable polymer. Meanwhile, the benzene ring and ether group in its polymer chain make it both rigid and flexible, leading to its excellent mechanical properties. In addition, the sulfone group and the whole structural unit form a large conjugate system, endowing it with high solvent resistance. In our previous study, sulfonated poly (aryl ether sulfone) (SPES) membranes were prepared by introducing sulfonic groups into the PES polymer chain and used for the separation of MeOH/ MTBE mixture [46]. The introduction of sulfonic groups enhanced the interaction between the membrane material and MeOH molecules and increased the proportion of amorphous regions in the membrane. The sulfonic groups interacted with each other to form sulfonic group clusters, building a molecular transport channel. The pervaporation performance was significantly enhanced. In order to tailor the microstructure and surface properties of the membrane, the method of incorporating MOFs into SPES polymer matrix was adopted to further improve the pervaporation separation performance. MOFs material ZIF-67 was selected for the following reasons: 1) ZIF-67 shows excellent hydrophilicity and preferential adsorption for MeOH over MTBE [47]; 2) ZIF-67 can be prepared at room temperature with simple process and lower membrane preparing cost should be anticipated.

In this research work, we attempted to study the effect of ZIF-67 on the PV efficiency by introducing it into SPES matrix. The morphological and chemical properties of the developed membranes were analyzed by means of SEM, atomic force microscope (AFM), water contact angle, thermogravimetric analysis (TGA), mechanical property testing and positron annihilation technique (PAT). Then, the effect of ZIF-67 loading on the PV separation of MeOH/MTBE mixtures was broadly studied.

## 2. Materials and Methods

### 2.1. Materials

2-methylimidazole, with purity of 98%, was purchased from Shanghai Yien Chemical Company, Shanghai, China. Cobalt chloride hexahydrate (with purity of 98%) was supplied by Sigma Aldrich, Shanghai, China. Polyvinylpyrrolidone, methanol, concentrated sulfuric acid and methyl tert-butyl ether were from Tianjin Kemio Chemical Reagent Company, Tianjin, China. N-methylpyrrolidone (purity 99%) was supplied by Shanghai Macklin Biochemical Technology Company, Shanghai, China. Poly (aryl ether sulfone) was from Jiangsu Xuzhou Engineering Plastics Factory, Xuzhou, China. Deionized water was made in the laboratory.

### 2.2. ZIF-67 Preparation

Quantities of 519 mg of cobalt chloride hexahydrate, 600 mg of polyvinylpyrrolidone and 2630 mg of 2-methylimidazole were dissolved in 80 mL of methanol and kept at room temperature for 12 h. Polyvinylpyrrolidone is a surfactant and used here as a stabilizer in the reaction system. It can also adjust the morphology of the nanoparticles. Then the reaction solution was centrifuged three times to remove the incompletely reactant. Finally, the product was dried at 80 °C for 4 h to obtain ZIF-67, which was bright purple.

### 2.3. Membrane Preparation

Please see the reference [46] for the preparation of SPES and SPES membranes. Take SPES/ZIF-67 mixed matrix membrane with ZIF-67 mass fraction of 5% as an example. The preparation process is as follows: 9.5 g of SPES was dissolved in 19 mL of N-methylpyrrolidone. After completely dissolved, 0.5 g of ZIF-67 was added into the solution and stirred violently for 30 min, followed by ultrasonic treatment for 3 times, 10 min for each time. The resultant solution was then left overnight to remove the bubbles. Then, 3 mL casting solution was casted on the clean glass plate, dried at 70 °C for 4 h, and then gently peeled off for further use. The obtained mixed matrix membrane was designed as M-x, where x represents the mass fraction of ZIF-67.

### 2.4. Membrane Characterization

Scanning electron microscopy (SEM, JSM-6490LV, JEOL, Kyushu Silicon Island, Japan) was used to characterize the micro-morphology of ZIF-67 and the mixed matrix membranes. The particle size distribution of ZIF-67 was measured with laser particle size analyzer (LDPA, Microtrac S3500, Michick, Michigan City, MI, USA). The crystal structure of ZIF-67 was characterized by X-ray diffraction (XRD, D8 ADVANCE, Bruker, Karlsruhe, Germany) using Cu Kα radiation. The diffraction was operated at 40 kV and 30 mA at the 2*θ* range of 5–40° using a step size of 0.0167° and a counting time of 10 s per step. The surface topography of the membranes was observed by an atomic force microscope (AFM, Nano-scope IIIa multimode SPM, Digital Instruments, Austin, TX, USA) with a commercial Si probe in a tapping model at 25 °C. The thermodynamic properties of the mixed matrix membranes were investigated by thermogravimetric analyzer (TGA, STA449F3, NETZSCH, Free State of Bavaria, Germany). Accurately weighted samples were placed into aluminum cups and heated from room temperature to 900 °C at a constant heating rate of 10 °C·min^−1^ under constant nitrogen purging at 10 mL·min^−1^. The mechanical properties of the membranes were tested by the electronic precision universal testing machine (AG-10KNIS MO, Shimadzu Company, Kyoto, Japan). The water contact angle measurements at room temperature were conducted using contact angle tester (SL200B, Shanghai Solon Information Technology Co., Ltd. Shanghai, China). The water contact angle at three different positions on the same membrane was measured for the average value. The free volume parameters of the membranes were determined by positron annihilation spectrometer (PALS, ORTEC fast-fast composite, EG&G Company, Boston, MA, USA).

### 2.5. Swelling and Sorption Experiments

Pre-dried membranes were weighed to obtain the mass of dry membrane (*m*_d_), and then immersed in MeOH (15 wt%)/MTBE mixtures at 40 °C for 48 h to reach an equilibrium swelling. During the process, the membranes were taken out at regular intervals, wiped with tissue paper carefully to remove the surface solvent, and weighed immediately, then dipped into the solution again. The procedure was repeated until the weight of the swollen membrane (*m*_w_) kept constant. For sorption behavior testing, the adsorbate was collected in a liquid nitrogen trap by desorbing the equilibrated sample in a purge-and-trap apparatus, and the concentration of the adsorbate was measured by gas chromatography. The testing was repeated three times and the average value was designed as the final. The degree of swelling (*DS*) and sorption selectivity (*α*_s_) were calculated as follows.
(1)$$DS(\%)=\frac{m_{w}-m_{d}}{m_{d}}\times 100$$
(2)$$\alpha_{S}=\frac{M_{MeOH}÷M_{MTBE}}{F_{MeOH}÷F_{MTBE}}$$
where *M_MeOH_* and *M_MTBE_* are the mass fraction of MeOH and MTBE in the membrane, *F_MeOH_* and *F_MTBE_* are that of MeOH and MTBE in the feed, respectively.

### 2.6. Pervaporation Performance Testing

The pervaporation separation performance of as-prepared membranes for MeOH/MTBE was investigated by using the pervaporation device (SULZER, Hamburg, Germany). The mass fraction of methanol in the feed was 15%. The feed flow rate was maintained at 50 L·h^−1^. The operating temperature and the pressure at the permeate side of membranes was kept at 40 °C and 400 Pa, respectively. The effective area of the membrane was 16.61 cm^2^. The permeant was collected in the liquid nitrogen cold trap after the unit has been running stably for 1 h. The mass of the permeant was weighed and then analyzed with gas chromatograph for its composition. The pervaporation performance testing for every membrane was repeated three times, and the average value was used as the final value. The total flux and separation factor were calculated by the following formula:(3)$$J=\frac{m}{A\times t}$$
(4)$$\alpha =\frac{Y_{i}÷Y_{j}}{X_{i}÷X_{j}}$$
where *J* is the total flux (kg·m^−2^·h^−1^), *m* is the weight of permeant (kg), *t* is the collecting time (h), *A* is the effective membrane area (m^2^), *α* is the separation factor, *X* and *Y* denote the mass fractions of component in the feed and the permeate, respectively.

## 3. Results and Discussion

### 3.1. ZIF-67 Characterization

Figure 1 shows the XRD spectrum and SEM image of ZIF-67. The SEM image demonstrated a shape of dodecahedral rhombus, which is a typical form of ZIF-67. The characteristic peaks corresponding to (011), (112) and (222) crystal faces were observed as shown in the XRD spectrum, which was consistent with the literature report [47]. This proves the successful synthesis of ZIF-67. Figure 2 shows particle size distribution of ZIF-67. The results showed that the particle size of ZIF-67 was between 180 nm and 580 nm, and the particle size was mainly in the range of 280 nm and 400 nm. This is beneficial to its dispersion in SPESC polymer matrix. As shown in Figure 3, the pore diameter of ZIF-67 was 0.51 nm.

### 3.2. Micromorphology of Mixed Matrix Membranes

Figure 4 shows the SEM images of all the membrane surface, cross-section of M-20 and its Co EDS mapping. Figure 4a showed that the surface of SPES film was smooth and free of defects. Figure 4b–e demonstrated that the compatibility between ZIF-67 and SPES matrix was good, and the membrane surface was uniform and dense without obvious defects. The introduction of ZIF-67 makes the membrane surface rougher. In addition, ZIF-67 particles were uniformly dispersed in SPES polymer matrix when ZIF-67 loading was lower than 20%. This was also confirmed by Co EDS mapping of M-20 in Figure 4h. Figure 4f exhibited that agglomeration phenomenon of ZIF-67 particles was observed when the mass fraction of ZIF-67 reached to 25%. Figure 4b,f also showed that part of ZIF-67 was exposed on the membrane surface and could directly interact with the permeable molecules, which would facilitate the selective adsorption of MeOH molecules on the membrane surface.

AFM diagrams including 2D images and 3D images of the mixed matrix membrane surface are given in Figure 5. It could be seen that the root mean square roughness of SPES membrane was 1.31 nm. When the mass fraction of ZIF-67 was 15%, the value of M-15 membrane increased to 6.45 nm. With further increasing the mass fraction of ZIF-67, the root mean square roughness value was as high as 8.45 nm. This can be mutually confirmed with SEM characterization results.

### 3.3. Mechanical Properties of MMMs

Figure 6 shows the mechanical properties of SPES membrane and the mixed matrix membranes. The results demonstrated that the tensile stress of all MMMs was higher than that of SPES membrane. With the mass fraction of ZIF-67 increasing from 5% to 25%, the tensile stress increased from 28.4 MPa to 36.9 MPa. However, the introduction of ZIF-67 reduced the strain of the membranes. Moreover, with the increase of the mass fraction of ZIF-67, the strain of the mixed matrix membranes gradually decreased from 106% to 54%. This is because the incorporation of ZIF-67 changes the stacking style of SPES polymer chains in the matrix, thus improving the rigidity of the mixed matrix membrane. The results are consistent with those reported in the literature [48].

### 3.4. TGA Analysis

Figure 7 shows the TGA curves of SPES membrane and mixed matrix membranes. Before 220 °C, the mass of the membranes decreased slowly, which was caused by the volatilization of residual solvent in the membrane matrix. This temperature is higher than the operating temperature of pervaporation process. Due to the thermal degradation of sulfonic groups in SPES, the membrane mass continued to decrease in the temperature range of 220 °C to 290 °C [47]. When the temperature was higher than 450 °C, the thermal degradation of SPES polymer backbone and ZIF-67 led to a rapid decline. This proves that the mixed matrix membranes meets the requirements of pervaporation operation.

### 3.5. Membrane Hydrophilicity and Degree of Swelling

MeOH/MTBE mixture is a typical polar/non-polar binary system. The hydrophilicity of MeOH-selective membrane is an important indicator. Figure 8 shows the effect of ZIF-67 mass fraction on the water contact angle of the membrane surface. It could be seen that the water contact angle of all membranes was lower than 90°, indicating the hydrophilicity of as-prepared membranes. The water contact angle of SPES film was 71.2°. With the increase of ZIF-67 mass fraction, the water contact angle decreased. The water contact angle of M-25 membrane was lowest as 60.7°. This is because the water contact angle of pure ZIF-67 membrane is 24° [47]. Its hydrophilicity is stronger than that of SPES. Some ZIF-67 particles are exposed on the mixed matrix membrane surface, thus improving the hydrophilicity.

To better illustrate the preferential sorption problem, we conducted the sorption and swelling experiments to discuss the preferential sorption of the separated components in the membrane matrix. The results are demonstrated in Figure 9. As can be seen, the degree of swelling of the membranes increased from 13.1% to 25.3% with increasing ZIF-67 loading. The sorption selectivity of SPES was 6.03. With increasing ZIF-67 loading from 5% to 25%, the sorption selectivity was enhanced from 6.5 to 8.4. This indicates that MeOH has stronger interaction with the membrane surface and can be preferentially adsorbed compared with MTBE. In addition, the interaction between MeOH and membrane surface can be strengthened by increasing ZIF-67 loading amount.

### 3.6. Free Vloume Parameters of MMMs

The free volume parameters of the membranes can be quantitatively analyzed by positron annihilation technology. The free volume parameters mainly include the free volume fraction and cavity radius of the membranes, which provides space for the diffusion of permeant molecules. The results are shown in Figure 10. The free volume fraction and cavity radius of SPES membrane were 8.37% and 0.271 nm, respectively. With ZIF-67 mass fraction increasing from 5% to 25%, the cavity radius of mixed matrix membranes increased from 0.289 nm to 0.324 nm, while the free volume fraction increased from 8.56% to 11.89%. The above results demonstrated that free volume fraction and cavity radius are positively correlated with ZIF-67 mass fraction. The reason for the above phenomenon is that ZIF-67 is uniformly dispersed in SPES-C polymer matrix, which changes the stacking style of SPES polymer chains. Meanwhile, the high porosity of ZIF-67 plays a positive role in improving the free volume parameters. This will provide more diffusion space for permeant molecules.

### 3.7. Pervaporation Performance of MMMs

Figure 11 exhibits the effect of ZIF-67 mass fraction on the pervaporation performance of mixed matrix membranes. The results showed that the separation factor of SPES membrane was 1157, which was lower than that of all mixed matrix membranes. With ZIF-67 mass fraction increasing from 5% to 20%, the separation factor increased from 1769 to 2123. When ZIF-67 mass fraction was 25%, the separation factor rapidly declined to 1435. This is because the cavity diameter (twice of the cavity radius) of M-20 membrane reaches 0.63 nm (see Figure 10), which is slightly higher than the molecular dynamics diameter of MTBE (0.62 nm). When ZIF-67 mass fraction is less than 20%, the preferential adsorption of MeOH and the diffusion resistance of MTBE in the membrane are conducive to the improvement of separation factor. When the mass fraction of ZIF-67 is further increased to 25%, the cavity diameter of M-25 membrane increases to 0.65 nm, resulting in a decrease in the diffusion resistance of MTBE in the membrane, which leads to competition with the diffusion of MeOH. Moreover, some agglomeration of ZIF-67 occurred with ZIF-67 loading as 25%, as shown in Figure 4. Around the cluster of aggregated particles, voids may form, and the selectivity of the membrane decreases. This also leads to the decline of separation factor. The total flux of SPES membrane is 0.178 kg·m^−2^·h^−1^. With the increase of ZIF-67 mass fraction, the total flux increased from 0.214 kg·m^−2^·h^−1^ to 0.321 kg·m^−2^·h^−1^. This can be explained from two aspects. Firstly, the increasing free volume fraction of mixed matrix membranes provides more mass transfer space for the permeable molecules (see Figure 10). Secondly, the moderate degree of swelling also benefits the enhancement of total flux. The above two aspects together play a positive role in the increase of total flux.

Above all, it is exciting to observe the anti-trade-off effect between total flux and separation factor when ZIF-67 mass fraction is lower than 20%. Both the improved *FFV* (Figure 10) and moderate swelling (Figure 9) contribute to the total flux enhancement. The solution-diffusion mechanism is usually applied to explain the separation factor of as-prepared membranes. The improved interaction between MeOH and membrane surface (Figure 8 and Figure 9) with embedding ZIF-67 favors the preferential adsorption of MeOH. In addition, the membranes with filler loading below 20 wt% exhibit a lower 2*R* value than the kinetic diameter of MTBE molecule (0.62 nm) (Figure 10), thereby MTBE molecule encounters higher transport resistance compared with MeOH molecule (the kinetic diameter of 0.40 nm) during the diffusion process. The above two aspects contribute to the reinforcement of separation factor. Pervaporation separation performance can be comprehensively evaluated by pervaporation separation index (PSI, kg·m^−2^·h^−1^), and the equation is PSI = *J*(*α*-1). As calculated, the PSI value of M-20 membrane was the highest as 630.234 kg·m^−2^·h^−1^, indicating that its comprehensive separation performance is the best.

### 3.8. Comparison of the Present Work and the Literature Data

A comparison of PV performance in this work with reported results is essential. The available data in the literature and results in this work are listed in Table 1. It can be seen that a wide range of separation factors from 6 to 4000 have been reported for different membranes. The separation factors of SPES/ZIF-67 mixed matrix membranes remain at a high level. Especially, the separation factor of M-20 reaches to 2123 for PV separation of 15 wt% MeOH feed mixture at 40 °C. CA/PVP, PHB, PLA and CS/surfactants membranes exhibit higher total flux than M-20, but separation factor is lower. Although PVA/PAA membrane shows the highest separation factor as 4000, its total flux is comparably low. A great PV performance improvement of SPES/ZIF-67 mixed matrix membranes for separation of MeOH/MTBE mixtures is found. It is reasonable to conclude that the MMMs as-prepared in this work have great potential for practical application.

## 4. Conclusions

ZIF-67 was successfully synthesized and introduced into sulfonated poly (aryl ether sulfone) (SPES) matrix to prepare a series of SPES/ZIF-67 mixed matrix membranes and used for pervaporation separation of MeOH/MTBE mixture. ZIF-67 showed good compatibility with SPES matrix and was uniformly dispersed in the mixed matrix membrane. The as-prepared mixed matrix membrane showed no defects. However, some agglomeration occurred when ZIF-67 loading was 25%. The introduction of ZIF-67 could meet the mechanical and thermal requirements of pervaporation process. ZIF-67 enhanced membrane surface hydrophilicity and preferentially adsorption of MeOH. The tune of free volume parameters was realized, which improved the pervaporation separation performance. When the ZIF-67 mass fraction was lower than 20%, the total flux and separation factor increased simultaneously, which exhibited the anti-trade-off effect. When the mass fraction of ZIF-67 is 20%, the mixed matrix membrane demonstrated the best pervaporation performance. The total flux and separation factor reached 0.297 kg·m^−2^·h^−1^ and 2123, respectively. In summary, the mass transfer of the penetrant molecules in the dense membranes for pervaporation is located in the scope of confined mass transfer. In this work, it is confirmed that incorporating nano-filler ZIF-67 into SPES polymer matrix to construct mixed matrix membranes (MMMs) is an effective approach to tune the confined mass transfer structure. This provides an effective way to enhance the pervaporation performance of the widely used polymeric membranes.

## Figures and Tables

**Figure 1 membranes-13-00389-f001:**
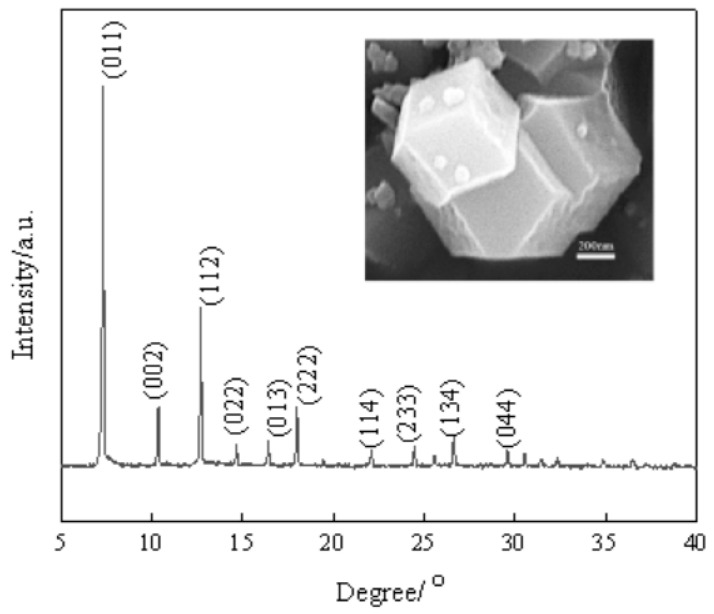
XRD pattern and SEM image of ZIF-67.

**Figure 2 membranes-13-00389-f002:**
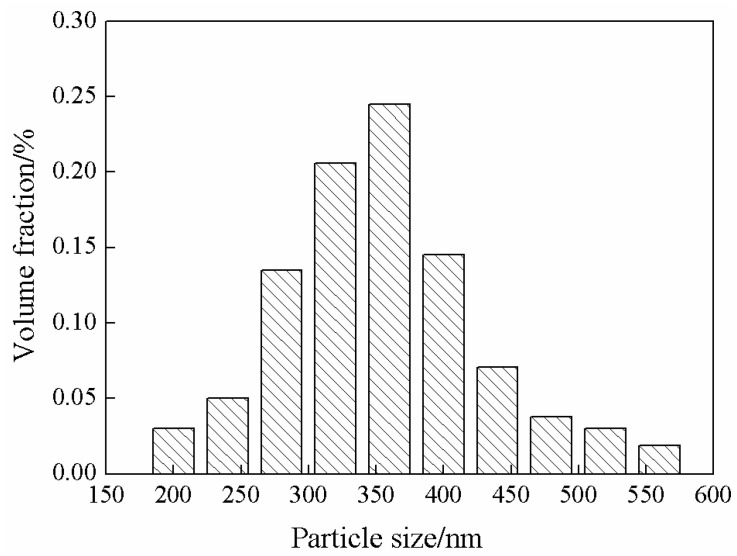
Particle size distribution of ZIF-67.

**Figure 3 membranes-13-00389-f003:**
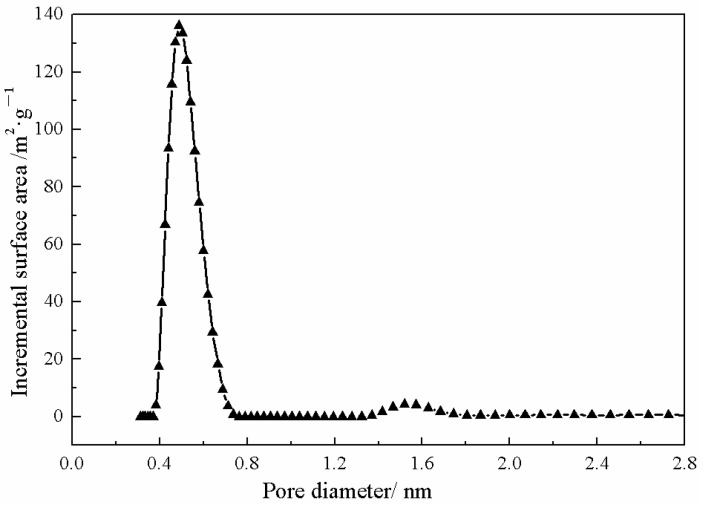
Pore size of as-prepared ZIF-67 particles.

**Figure 4 membranes-13-00389-f004:**
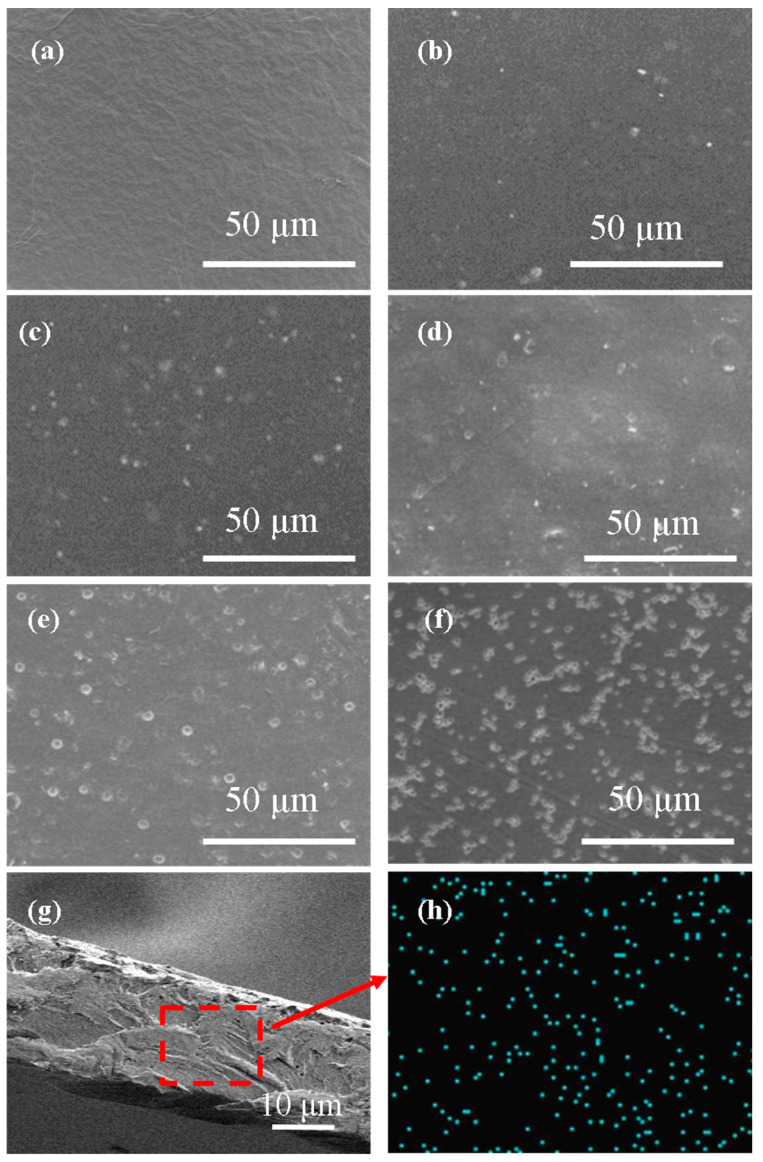
SEM images of the mixed matrix membranes (membrane surface of (**a**) M-0, (**b**) M-5, (**c**) M-10, (**d**) M-15, (**e**) M-20, (**f**) M-25, (**g**) M-20 cross-section and (**h**) Co EDS mapping).

**Figure 5 membranes-13-00389-f005:**
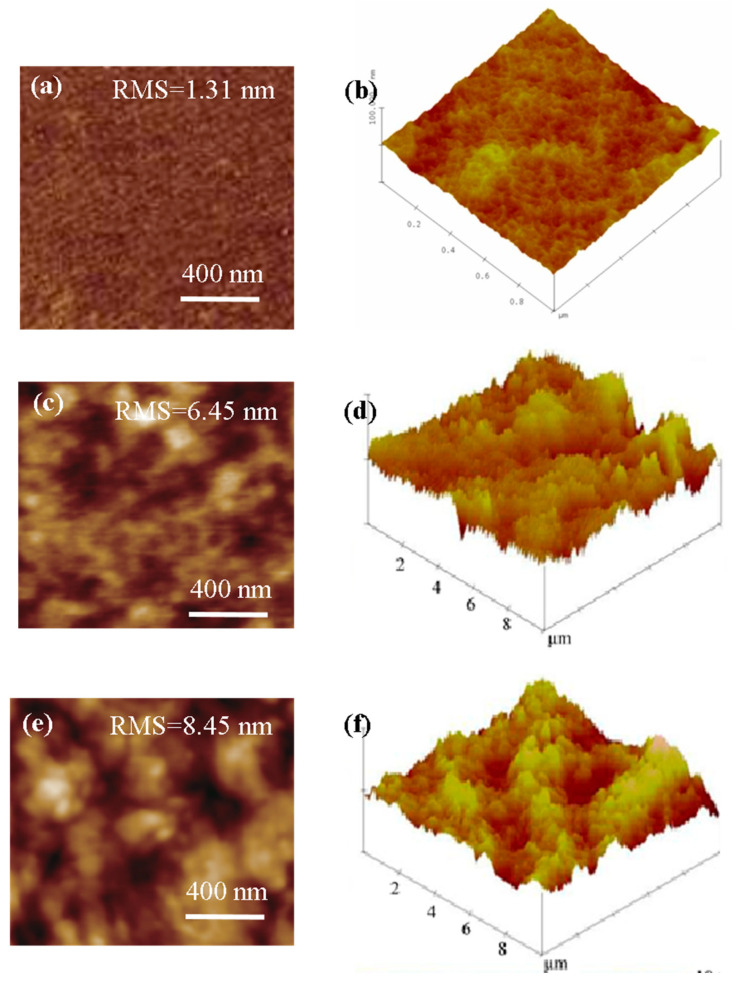
AFM images of the mixed matrix membranes (2D images and 3D images of membrane surface of (**a**,**b**) M-0, (**c**,**d**) M-15 and (**e**,**f**) M-25).

**Figure 6 membranes-13-00389-f006:**
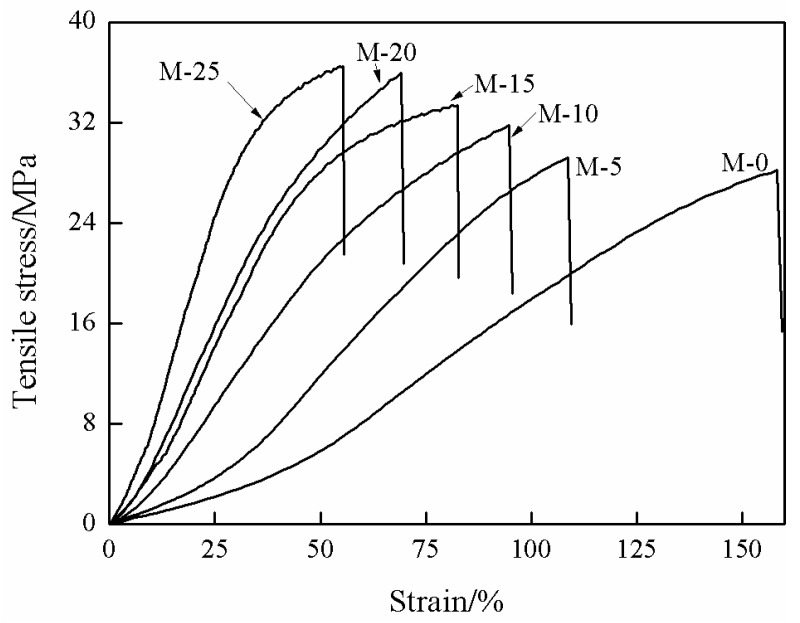
Mechanical properties of the mixed matrix membranes.

**Figure 7 membranes-13-00389-f007:**
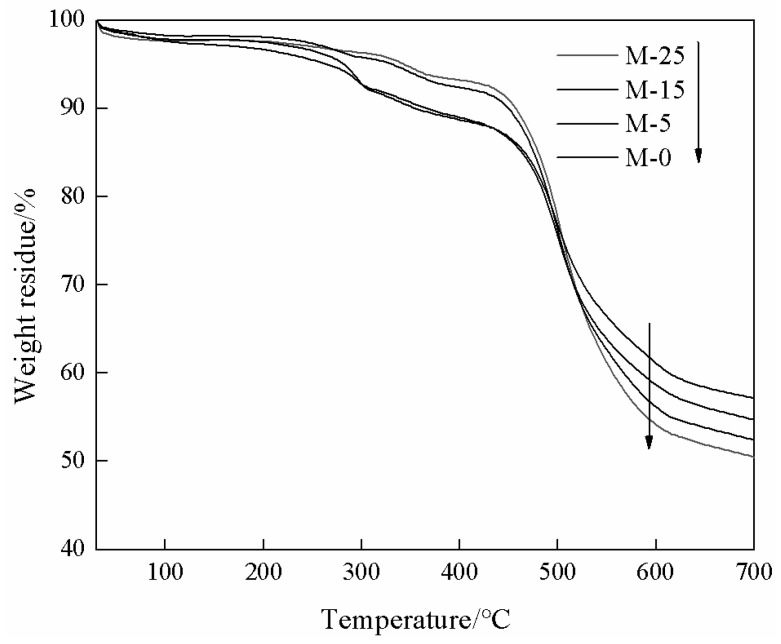
TGA curves of the mixed matrix membranes.

**Figure 8 membranes-13-00389-f008:**
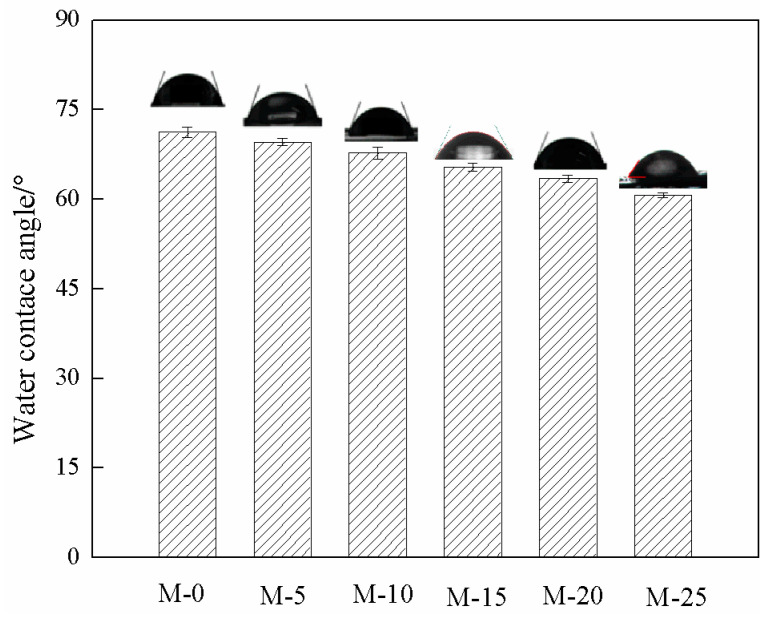
Water contact angle of mixed matrix membranes.

**Figure 9 membranes-13-00389-f009:**
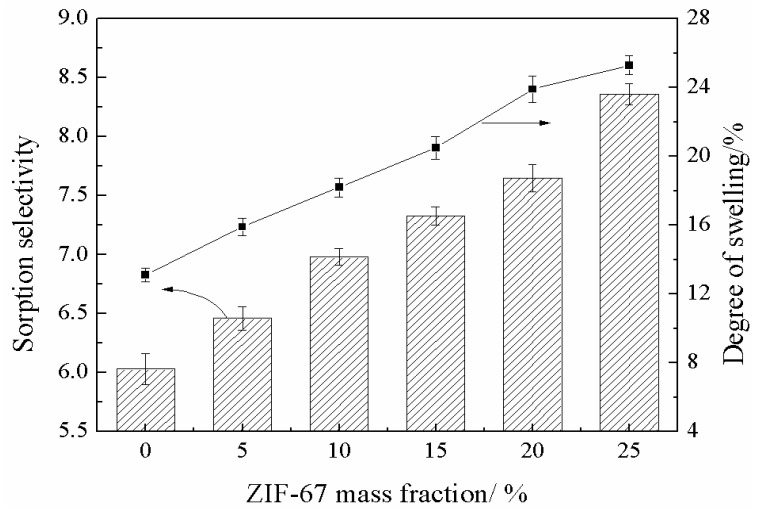
Sorption selectivity and degree of swelling of mixed matrix membranes.

**Figure 10 membranes-13-00389-f010:**
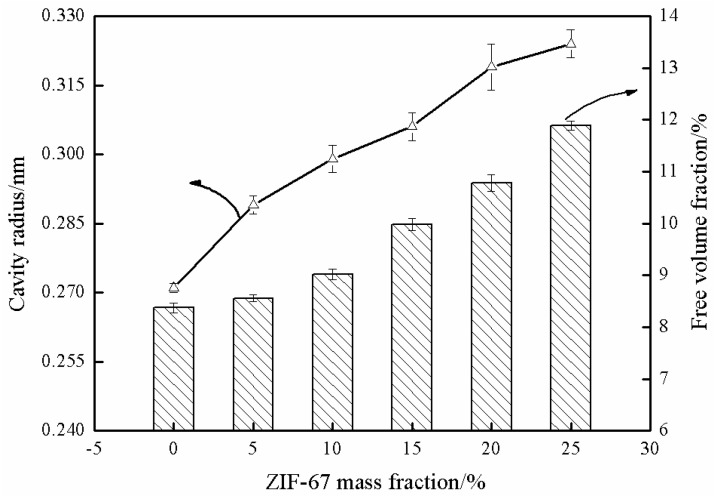
Free volume parameters of mixed matrix membranes.

**Figure 11 membranes-13-00389-f011:**
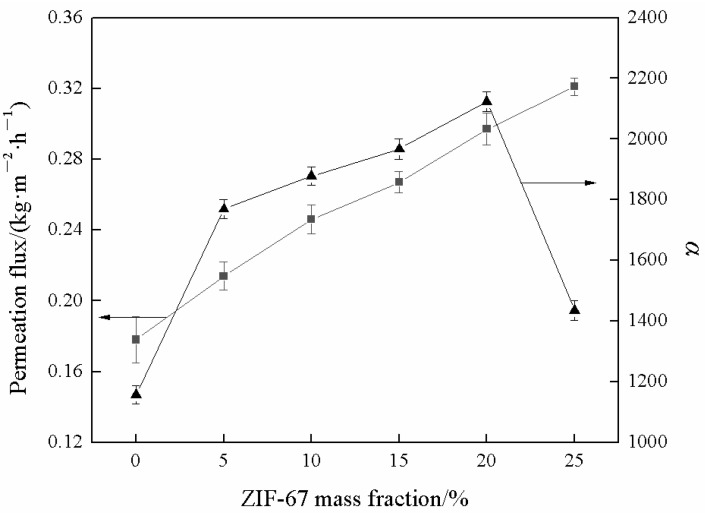
Effect of ZIF-67 mass fraction on pervaporation performance of mixed matrix membranes.

**Table 1 membranes-13-00389-t001:** PV performance comparison of MMMs in this work and in literature.

Membranes	MeOH in Feed	Temperature (°C)	*J* (kg·m^−2^·h^−1^)	*α*	Reference
HZSM5/CA	20 wt%	30	0.226	346	[48]
CA/PVP	20 wt%	40	0.430	411	[49]
PEEK-C	20 wt%	20	0.050	15	[50]
PHB	40 mol%	40	0.387	9	[51]
PLA	20 wt%	30	0.552	6	[52]
PES-C	40 wt%	40	0.108	99	[53]
PVA/PAA	15 wt%	50	0.010	4000	[54]
CS/surfactants	20 wt%	25	0.653	231	[55]
CA/ZnO	31 wt%	40	0.115	760	[56]
SPES-C/PEI	15 wt%	40	0.194	1860	[57]
PES-C/PVP	15 wt%	40	0.172	889	[58]
M-20	15 wt%	40	0.297	2123	This work

Abbreviations: CA, cellulose acetate; CS, chitosan; PAA, poly (acrylic acid); PEEK-C, poly (ether ether ketone) with cardo; PEI, polyethyleneimine; PES-C, polyaryl ether sulfone with cardo; PHB, poly(3-hydroxybutyrate); PLA, poly(lactic acid); PVA, polyvinyl alcohol; PVP, polyvinyl pyrrolidone.

## Data Availability

The data will be supplied as required.

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
