# Peer review of "ZIF-67 Incorporated Sulfonated Poly (Aryl Ether Sulfone) Mixed Matrix Membranes for Pervaporation Separation of Methanol/Methyl Tert-Butyl Ether Mixture"

_membranes, 2023, doi:10.3390/membranes13040389_

Round 1

Author Response

Thank you for your valuable comments.

Reviewer 2 Report

This manuscript reported a series of poly(aryl ether sulfone) membranes modified by ZIF-67.  The efficiency of the membranes was evaluated in the separation of the methanol/methyl tert butyl ether mixture by pevaporation. The contents and results are of great interest to the membrane community, while some significant flaws exist in the discussion and introduction.

1.  Could authors provide information on why poly (vinylpyrrolidone) was used during the synthesis of the ZIF-67?

2. Could authors provide the SEM images for other membranes? Moreover, it should be better if the authors will provide the SEM micrographs with a smaller approximation (e.g. 100 micrometres). Now we can not say it the membranes possess agglomeration or not.

4. It would be easier to see the effect of the incorporation of the ZIF-67 into the membrane matrix on surface roughness if the 3D image will be added to the manuscript. 

5. Lower contact angle of water than 90 degrees does not indicate that the membrane has a stronger interaction with the surface of the membrane. It's only can say that the membrane is hydrophilic.  In the case of the preferential sorption of the separated components in the matrix of the membrane, authors should calculate the distance parameter based on the Hansen Solubility Parameters. Based on the calculation it can be predicted which compound could has stronger interaction during the sorption.

6. Line 206. Theoretically, if the surface roughness of the membrane increased the contact angle of water also should be increased (especially in this case, where surface roughness increased from 1.31 nm to 8.45). But it can say they that an increase in surface roughness leads to an increase in the hydrophilic character of MMMs. Hydrophilicity depends on the character of the materials, not the roughness. So I suggest to the authors calculate the surface free energy to actually proved that incorporation of the ZIF-67 increases the hydrophilic character of the membranes.

7. Could the author provide the pore size of the ZIF-67? and MTBE? In the discussion author only used the cavity radius but in my opinion, it is also important to add information about the pore of filler. For now, we don't know if we have the sharp sieve effect and MTBE is rejected or maybe the kinetic diameter of ZIF-67 is higher (or much higher) compared with both MTBE and MeOH (in this case the interaction between ZIF-67 and the separated component will play a key role in the separation). Additionally, the authors should take into account the effect of membrane swelling during the evaporation process.

8. Based on the results presented in Figure 9, it can be also concluded that some agglomeration of ZIF-67 may occur. At the high content of filler, usually, there are some agglomerates. Around the cluster of aggregated particles, voids may form, and the selectivity of the membrane decreases with simultaneously increasing the transport properties. 

9. Could the authors provide a comparison with literature data? 

Author Response

Thank you for your suggestions.

Round 2

Reviewer 1 Report

Now, after a revision, paper is suitable for publication in membranes